# Health professionals' attitudes towards traditional healing for mental illness: A systematic review protocol

**Alemayehu Molla Wollie** [1,2]*, **Kim Usher**[1], **Reshin Maharaj**[1], **Md Shahidul Islam**[1]

1 Faculty of Medicine and Health, School of Health, University of New England, Armidale, NSW, Australia,
2 Department of Psychiatry, College of Medicine and Health Sciences, Injibara University, Injibara, Ethiopia

* alexmolla09@gmail.com

**Data Availability Statement:** All data are in the manuscript.

**Funding:** The author(s) received no specific funding for this work.

## Abstract

### Background

Mental illness is a global problem that receives less attention, particularly in developing countries. Integrating modern treatment with traditional healing approaches has been proposed as one way to address mental health problems, especially in developing countries. Despite health professionals' participation in traditional healing being crucial to integrative approaches, their participation is limited to date. This review protocol is designed to explore the attitudes of health professionals towards traditional healing practices in mental health services.

### Methods

The review will follow the Preferred Reporting Items for Systematic Review and Meta-Analysis (PRISMA) guidelines. Searching databases, including PubMed/Medline, PsychINFO, EMBASE, Scopus, and the Web of sciences will be conducted. Additionally, Google and Google Scholar will be searched for other information, including grey literature. Moreover, a manual search of identified articles' reference lists will also be conducted to help ensure all potential papers are included in the review. Qualitative, quantitative, and mixed study methods published in English between January 2014 and April 2024 will be included. The qualities of the included studies will be assessed using the Mixed Methods Appraisal Tool (MMAT) Version 2018. A mixed-method synthesis will be used to synthesis the results.

### Discussion

It is crucial for healthcare professionals to provide culturally sensitive care to empower people to manage their health. This systematic review will summarize the attitudes of health professionals towards the adoption and delivery of traditional healing approaches to people experiencing mental illness. Therefore, the findings of this review will support integration between traditional healers and modern mental health practitioners for the treatment of mental illness.

**Competing interests:** The authors have declared that co competing interest exists.

## Trial registration

**Protocol registration number:** CRD42024535136.

## Background

Mental health is the overall wellbeing and functioning of an individual, family, and community [1,2]. It is "a state of wellbeing in which the individual realizes his or her own abilities, is able to contribute to his or her environment, copes with the normal stress of life, works fruitfully, and is able to cooperate with others [3]." In contrast, mental illness has a substantial impact on a person's feelings, thoughts, behaviour, and social interactions [4,5]. It is influenced by numerous factors, including psychological, physical, social, cultural, and spiritual [6,7].

Currently, mental illness is a major public health burden and accounts for 32.4% of years lived with disability worldwide [8]. Even though mental health issues are a global health concern, they are highly prevalent in low-income countries [9,10]. Developing nations have fewer health professionals [11], resulting in a treatment gap for people with mental illnesses in these countries [12]. Furthermore, negative attitudes, stigma, limited resources, and low priority are reasons for the high magnitude of mental illness, in addition to a shortage of health professionals [13–16].

People with mental health issues often seek care from indigenous or traditional healers [17–19]. Traditional healing practices include but are not limited to a set of beliefs that use culturally accepted spiritual treatments and plant products [20]. Traditional healers form a major part of the mental health workforce in developing countries [21]. Nearly 80% of people from Africa seek medical attention from traditional healers [21], largely because traditional healers provide culturally and socially accepted care for individuals and communities. This is the preferred treatment approach for most people due to its perceived effectiveness, affordability, and accessibility within communities [22–24]. In order to close the significant treatment gap in low-and middle-income countries, "the World Health Organization's (WHO) 2003–2020 Mental Health Action Plan recommended that government health programs incorporate traditional healers as treatment resources [25]."

Studies indicate that traditional healers can offer efficacious therapies that may be beneficial for prevalent mental illnesses such as depression and anxiety [26,27]. Healers usually come from extended family branches with experience providing care like counselling, bible interpretation, and using prayer aids such as holy water and oil for treatment [27,28]. They either obtain experience through on-the-job training or apply their practices through ancestors who serve as mediators by providing access to spiritual guidance and power [15,29,30].

### Rationale

Despite the fact that traditional healing approaches are practiced by large groups of people, particularly in low-income countries, modern health professionals tend to differ in their opinions about this approach [31,32]. Studies indicate that some health professionals hold a positive view of traditional healing modalities [33], while many others express doubt about the effectiveness of traditional treatments. Furthermore, there is a negative attitude among health professionals regarding the integration of traditional healing with biomedical therapies [34,35]. Traditional healers are known to consult with modern health professionals when faced with problems beyond their ability to treat, but it is not common for biomedical professionals to refer back to healers [27]. This is often due to their negative attitudes toward

traditional healing practices, including the belief that traditional healers do not incorporate a human right approach and/or compassionate care when dealing with people with mental health problems [35,36]. Their negative attitudes have a detrimental effect on the WHO's recommended collaboration of biomedical treatment and traditional healing to close the treatment gap in low and middle-income countries because cooperation between the two needs mutual respect and recognition. In addition to providing biological therapies, it is good to emphasize spiritual and cultural approaches for mental illness since mental illness is due to multidimensional factors like spiritual, social, psychological, and physical. Moreover, it is crucial for healthcare providers to understand their patients' cultural beliefs and practices in order to manage and counsel them appropriately [34] and empower them towards healthful healing. There is no summarized research regarding the attitudes of health professionals toward traditional healing approaches for mental illness, despite the existence of numerous singe studies. Therefor, this review protocol is aimed to explore the summarized evidences on health care providers' attitudes toward traditional healing for mental illness, and this will be crucial in developing appropriate guidelines for holistic treatment.

## Research question

What are the health professionals' attitudes towards traditional healing for mental illness?

## Methods and materials

The Preferred Reporting Items for Systematic Review and Meta-Analysis (PRISMA) 2020 guidelines will be used to conduct this review [37]. This protocol is also prepared in accordance with the PRISAM-Protocol guidelines (Supporting Information). Primary studies conducted using quantitative, quantitative, and mixed-methods will be included. The review protocol is registered with the International Prospective Register of Systematic Reviews (PROSPERO) with a registration number of CRD42014535136.

## Eligibility criteria

The eligibility assessment format will be used to select articles to be included in the systematic review. Globally published original articles focusing on health professionals' attitude and/or perception towards traditional healing of mental illness will be considered. Qualitative, quantitative, and mixed-method studies will be considered without restrictions in their study design. Studies published in English between January 2014 and April 2024 will be considered to summarize contemporary evidence on the topic. Conference summaries, reviews, dissertations, case studies, governmental and non-government reports, and letters will be excluded from the study.

## Searching strategy

A literature search of databases such as PubMed/Medline, PsychINFO, EMBASE, Scopus, and the Web of Sciences will be conducted. The key terms will be searched by connecting Boolean operators OR/AND to specifically address published studies. Searching words will be (attitude) OR (perception) OR (belief) OR (opinion) OR (view) AND (health professionals) OR (health practitioners) OR (health personnel) OR (nurses) OR (medical doctors) OR (psychiatrist) OR (psychologist) AND (traditional healing) OR (traditional medicine) OR (herbal medicine) OR (indigenous treatment) OR (spiritual therapy) OR (religious healing) AND (mental illness) OR (mental disorder) OR (mental health) OR (psychological distress) OR (psychiatric disorder). In addition to database and manual searches, we will use references to articles obtained

from the database to filter the remaining studies. Database searching and the initial screening process will be finalized in consultation with a senior health librarian from the University of New England (UNE) Dixon Library and other authors.

An example of a search strategy for Scopus ( ( ALL ("health professional") OR TITLE-ABS-KEY ("health personnel") OR TITLE-ABS-KEY ("medical doctor") OR TITLE-ABS-KEY ("medical practitioner") OR TITLE-ABS-KEY (nurse) OR TITLE-ABS-KEY (psychiatrist) OR TITLE-ABS-KEY (psychologist) ) ) AND ( ( ALL (perception) OR TITLE-ABS-KEY (attitude) OR TITLE-ABS-KEY (belief) OR TITLE-ABS-KEY (view) OR TITLE-ABS-KEY (opinion) ) ) AND ( ( ALL ("traditional healing") OR TITLE-ABS-KEY ("traditional medicine") OR TITLE-ABS-KEY ("indigenous treatment") OR TITLE-ABS-KEY ("spiritual therapy") OR TITLE-ABS-KEY ("religious healing") OR TITLE-ABS-KEY ("herbal medicine") ) ) AND ( ( ALL ("mental illness") OR TITLE-ABS-KEY ("mental disorder") OR TITLE-ABS-KEY ("mental health") OR TITLE-ABS-KEY ("psychiatric disorder") OR TITLE-ABS-KEY ("psychological distress") ) ) AND PUBYEAR > 2013 AND PUBYEAR < 2025 AND (LIMIT-TO (DOCTYPE, "ar") ) AND (LIMIT-TO (LANGUAGE, "English") )

## Study selection process

PRISMA (preferred reporting items for the systematic review and meta-analysis) guidelines [37] will be followed to ensure the quality and transparency of the review process. After critically searching articles from databases, we will start the selection process by importing all records to an Endnote library. Duplicates will be detected and removed automatically. At the beginning, the selection process will be carried out by the first author (AMW), and it will be checked by one of the co-authors (KU, RM, and SI). The reviewer will evaluate article titles and abstracts for relevance, keeping those that are relevant to the outcome variable attitudes, and/or perceptions of health professionals towards traditional healing. The full texts of possible relevant papers will then be screened for eligibility. Any disagreements will be handled through a discussion to finalize the selection of the papers.

## Data extraction or collection process

With the use of a predetermined, uniform data extraction structure, the first author will extract important data, and then it will be confirmed by another author. Any significant differences will be discussed, and if a consensus cannot be reached, a third author will adjudicate on the decision. The first author's name, the year of publication, the country where the study is conducted, the methodology used, the health professional type, the data collection and analysis method, the major findings, and the conclusions will be included in the standardized data extraction form.

## Quality assessment

Each study's methodological quality and bias risk will be assessed to determine the validity of the findings. The qualities of the included articles will be assessed using the Mixed Methods Appraisal Tool (MMAT) Version 2018 Checklist [38]. This tool is designed to assess the methodological qualities of different types of studies, like qualitative, quantitative, and mixed studies. The differences in quality evaluation scores will be resolved by through discussion within a team.

## Data synthesis

An integrated mixed-method synthesis technique will be used to summarize all quantitative, qualitative, and mixed-method data in one combined synthesis [39]. Quantitative data will be

described qualitatively to facilitate integration with qualitative data. The overall nature of the articles will be presented in a table. Themes and subthemes will be formulated based on the data generated. All the data will be analysed and reported as a narrative summary. To minimize bias, the research team will thoroughly evaluate and discuss the entire process, and any disagreements will be resolved through discussion.

## Discussion

The review will be conducted to summarize the attitudes of health professionals towards traditional healing approaches for mental illness in a global context. Even though independent studies have been undertaken on health professionals' attitudes and/or perceptions towards traditional healing, there is no globally summarized general evidence on the topic. Some previous studies present opposing evidence on the attitudes of health professionals towards traditional healing practices. Even if some health professionals encourage traditional methods, there are other studies that show negative attitudes of health professionals towards traditional healing [34,35]. Traditional healing practices are considered appropriate options for mental illness, particularly in low-income countries where there is a scarcity of modern mental health professionals [21]. Most people with mental health problems seek these treatments because they provide culturally and socially accepted care for individuals and the community [22–24]. But there is limited collaboration between modern treatment and traditional healing. Therefore, this review will summarize all the literature and clearly show the positions of health professionals toward traditional healing approaches. The findings of this review will be important in developing directions for governmental organizations and other concerned bodies to strengthen communication and integration of traditional healing with biomedical treatment options for mental illness.

## Expected limitations

This systematic review has the limitation of excluding articles that are not published in English. In addition, the exclusion of conference reports, letter, short communications, and review articles may be a reason to miss important data.

## Publication status

This protocol is prepared for the ongoing systematic review of articles entitled "The health professionals' attitude towards traditional healing approaches to mental illness." If there is any change during the review, an updated version will be published with a final systematic review.

## Supporting information

**S1 Checklist. PRISMA-Protocol guidelines.**
(DOCX)

## Author Contributions

**Conceptualization:** Alemayehu Molla Wollie.

**Methodology:** Alemayehu Molla Wollie.

**Supervision:** Kim Usher, Reshin Maharaj, Md Shahidul Islam.

**Validation:** Kim Usher, Reshin Maharaj, Md Shahidul Islam.

**Visualization:** Kim Usher, Reshin Maharaj, Md Shahidul Islam.

**Writing – original draft:** Alemayehu Molla Wollie.

**Writing – review & editing:** Alemayehu Molla Wollie, Kim Usher, Reshin Maharaj, Md Shahidul Islam.

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
