## [Decision Letter · Decision Letter 0]

23 Jul 2024

PONE-D-24-18093Perceptions of health professionals towards traditional healing for mental illness: A systematic review ProtocolPLOS ONE

Dear Dr. Wollie,

Thank you for submitting your manuscript to PLOS ONE. After careful consideration, we feel that it has merit but does not fully meet PLOS ONE’s publication criteria as it currently stands. Therefore, we invite you to submit a revised version of the manuscript that addresses the points raised during the review process.

We look forward to receiving your revised manuscript.

Kind regards,

Madhulika Sahoo, Ph.D

Academic Editor

PLOS ONE

Journal Requirements:

"All authors declare that there is no conflict of interest regarding this work."

Reviewers' comments:

Reviewer's Responses to Questions

**Comments to the Author**

1. Does the manuscript provide a valid rationale for the proposed study, with clearly identified and justified research questions?

Reviewer #1: Yes

Reviewer #2: Partly

2. Is the protocol technically sound and planned in a manner that will lead to a meaningful outcome and allow testing the stated hypotheses?

Reviewer #1: Yes

Reviewer #2: No

3. Is the methodology feasible and described in sufficient detail to allow the work to be replicable?

Reviewer #1: Yes

Reviewer #2: No

4. Have the authors described where all data underlying the findings will be made available when the study is complete?

Reviewer #1: Yes

Reviewer #2: No

5. Is the manuscript presented in an intelligible fashion and written in standard English?

Reviewer #1: Yes

Reviewer #2: No

6. Review Comments to the Author

You may also provide optional suggestions and comments to authors that they might find helpful in planning their study.

Reviewer #1: The research will address an important area covering traditional healing practices for mental health illness.

Reviewer #2: 1. The article needs to be proofread and might require a thorough review of its English language usage to enhance clarity and readability.

2. The article overall lacks clarity on the relevance even though the protocol mentioned is good.

3. The article is missing on the novelty of the subject.

7. PLOS authors have the option to publish the peer review history of their article (what does this mean?). If published, this will include your full peer review and any attached files.

Reviewer #1: **Yes: **Dr. Aradhana Panigrahi

Reviewer #2: No

---

## [Author Response · Author response to Decision Letter 0]

31 Jul 2024

Response to reviewers 

“PONE-D-24-18093: Perceptions of health professionals towards traditional healing for mental illness: A systematic review protocol”

Dear editor, 

Thank you very much for giving us the chance to revise this study protocol. We have tried to update the study protocol and are happy to write this response letter to comments.

Reviewers comments

Reviewer #1:

Comment: “The research will address an important area covering traditional healing perspectives for mental illness.”

Response: 

Dear reviewer #1, 

We sincerely appreciate you taking the time to provide feedback on this study protocol. As you mentioned, we have gathered large numbers of studies and will try to publish the full systematic review soon, which will include significant summarized data and provide direction to decision-makers and other interested parties. 

Reviewer #2:

Comment: “The article needs to be proofread and might require thorough review of its English language usage to enhance clarity and readability.” 

Response: 

Dear reviewer #2,

We greatly appreciate your significant input. We have proofread the entire study protocol and rewritten some sections of it in accordance with your comment. We hope the updated version is clear and understandable.

Comment: “The article overall lacks clarity on the relevance even though the protocol mentioned is good”

Response: Regarding the relevance of the study, we have attempted to mention the importance or rationale for our plan on page 2 of this protocol. In addition, please consider the following issues regarding the necessity of doing a systematic review on this topic and the outcome after this protocol: 

First, some biomedical professionals are against the traditional healing approaches and collaborative work, even though the WHO recommends them to bridge the treatment gaps for mental illness. These attitudinal problems are obstacles to counselling and empowering patients towards a good recovery process. This is due to the possibility that, in the absence of mutual communication, transparency, and task sharing with healers, they may be less able to comprehend cultural understandings and widely utilized community healing techniques. Improved communication and collaboration between health professionals and traditional healers are necessary to provide holistic treatment for mental illness. 

Second, unless there is effective and open communication, separately delivered treatment approaches may pose risks because patients may have a chance of developing adverse effects while mixing medications from two treatment modalities without clear consultation. 

Lastly, this protocol is planned to produce summarized evidence incorporating globally conducted studies on health professionals’ attitudes and/or perceptions toward traditional healing for mental illness. This is highly pertinent to many stakeholders in providing comprehensive and cost-effective mental health care.

Comment: “The article is missing on the novelty of the subject.”

Response: Regarding the novelty of the study, we initially carried out a detailed search to locate related studies. But, as per our knowledge and search strategy, we were unable to identify globally published summarized review articles on health professionals’ attitudes and/or perceptions toward traditional healing approaches for mental illness. Following thorough searching, this protocol is then registered with the International Prospective Register of Systematic Reviews (PROSPERO) with a registration number CRD42014535136. After submitting this protocol to the journal, we have collected globally published single studies, and we have found important contemporary evidence for our final systematic review. 

If further revision is needed, we are happy to update it.

Kind regards!

Mr. Alemayehu Molla Wollie alexmolla09@gmail.com

Professor Kim Usher: kusher@une.edu.au

Dr. Reshin Maharaj: Reshin.Maharaj@une.edu.au

Dr. Md Shahidul Islam: mislam27@une.edu.au

---

## [Decision Letter · Decision Letter 1]

28 Aug 2024

Health professionals’ attitudes towards traditional healing for mental illness: A systematic review Protocol

PONE-D-24-18093R1

Dear Dr. Wollie ,

We’re pleased to inform you that your manuscript has been judged scientifically suitable for publication and will be formally accepted for publication once it meets all outstanding technical requirements.

Kind regards,

Madhulika Sahoo, Ph.D

Academic Editor

PLOS ONE

Additional Editor Comments (optional):

Reviewers' comments:

Reviewer's Responses to Questions

**Comments to the Author**

1. Does the manuscript provide a valid rationale for the proposed study, with clearly identified and justified research questions?

Reviewer #2: Yes

2. Is the protocol technically sound and planned in a manner that will lead to a meaningful outcome and allow testing the stated hypotheses?

Reviewer #2: Yes

3. Is the methodology feasible and described in sufficient detail to allow the work to be replicable?

Reviewer #2: Yes

4. Have the authors described where all data underlying the findings will be made available when the study is complete?

Reviewer #2: Yes

5. Is the manuscript presented in an intelligible fashion and written in standard English?

Reviewer #2: Yes

6. Review Comments to the Author

You may also provide optional suggestions and comments to authors that they might find helpful in planning their study.

Reviewer #2: The necessary changes have been done in revision 1. The authors have clearly answered to all the concerns raised in the 1st revision. The manuscript can be accepted in the revised form.

7. PLOS authors have the option to publish the peer review history of their article (what does this mean?). If published, this will include your full peer review and any attached files.

Reviewer #2: **Yes: **Salini Rosaline
